# Distance of Biopsy-Confirmed High-Risk Breast Lesion from Concurrently Identified Breast Malignancy Associated with Risk of Carcinoma at the High-Risk Lesion Site

**DOI:** 10.3390/cancers16122268

**Published:** 2024-06-19

**Authors:** Julie Le, Thomas J. O’Keefe, Sohini Khan, Sara M. Grossi, Hye Young Choi, Haydee Ojeda-Fournier, Ava Armani, Anne M. Wallace, Sarah L. Blair

**Affiliations:** 1Division of Breast Surgery, The Comprehensive Breast Health Center, University of California San Diego, 3855 Health Sciences Dr, La Jolla, CA 92093, USA; 2Department of Surgery, Jennifer Moreno Department of Veterans Affairs Medical Center, La Jolla, CA 92161, USA; 3Department of Surgery, University of California San Diego, 3855 Health Sciences Dr, La Jolla, CA 92093, USA; 4Division of Breast Imaging, University of California San Diego, 3855 Health Sciences Dr, La Jolla, CA 92093, USA; 5Department of Medicine, Gyeongsang National University College of Medicine, Jinju-si 52727, Republic of Korea

**Keywords:** breast cancer, high-risk breast lesion, synchronous breast cancer, intraductal papilloma, complex sclerosing lesion of breast, lobular hyperplasia, flat epithelial atypia

## Abstract

**Simple Summary:**

Incidental intraductal papilloma without atypia (IPA), lobular hyperplasia (LCIS or ALH), flat epithelial atypia (FEA) and complex sclerosing lesion (CSL) are high-risk lesions of the breast. These lesions are not routinely excised when diagnosed in isolation because they have low rates of upgrade to invasive cancer. When identified concurrently with invasive breast cancer, however, the upgrade rate is not well characterized. We sought to both characterize the upgrade rate for these lesions when diagnosed concurrently with invasive breast cancer, and to identify features of the high-risk lesions predictive of upgrade to cancer. In our cohort of 65 patients who were concurrently diagnosed with a high-risk breast lesion and an invasive cancer, 5 patients (7.7%) had an upgrade of their high-risk lesion to carcinoma. The rate of upgrade was higher for high-risk lesions that were ipsilateral to malignancy and within 5 cm of it, and lower for other lesions.

**Abstract:**

High-risk breast lesions including incidental intraductal papilloma without atypia (IPA), lobular hyperplasia (LCIS or ALH), flat epithelial atypia (FEA) and complex sclerosing lesion (CSL) are not routinely excised due to low upgrade rates to carcinoma. We aim to identify features of these lesions predictive of upgrade when identified concurrently with invasive disease. Methods: A single-center retrospective cohort study was performed for patients who underwent multi-site lumpectomies with invasive disease at one site and a high-risk lesion at another site between 2006 and 2021. A multinomial logistic regression was performed. Results: Sixty-five patients met the inclusion criteria. Four patients (6.2%) had an upgrade to in situ disease (DCIS) and one (1.5%) to invasive carcinoma. Three upgraded high-risk lesions were ipsilateral to the concurrent carcinoma and two were contralateral. In the multivariate model, a high-risk lesion within 5 cm of an ipsilateral malignancy was associated with increased risk of upgrade. The 3.8% upgrade rate for high-risk lesions located greater than 5 cm from ipsilateral malignancy or in the contralateral breast suggests that omission of excisional biopsy may be considered. Excisional biopsy of lesions within 5 cm of ipsilateral malignancy is recommended given the 25% upgrade risk in our series.

## 1. Introduction

The management of incidentally detected high-risk breast lesions including intraductal papilloma without atypia (IPA), classic lobular carcinoma in situ (cLCIS), atypical lobular hyperplasia (ALH), flat epithelial atypia (FEA) and complex sclerosing lesion (CSL) is evolving. Traditionally, excision was recommended for these high-risk lesions due to the highly variable upgrade rates to carcinoma reported when such lesions are excised [1,2,3,4]. More recently, upgrade rates of less than 5% have been reported for these lesions, and in 2016, the American Society of Breast Surgeons (ASBrS) consensus guidelines suggested that close surveillance and follow-up can be considered as an alternative to surgical excision [5].

### 1.1. Intraductal Papilloma

IPA is a lesion characterized histopathologically by a pedunculated intraductal mass on a fibrovascular stalk, and represents the most common benign cause of bloody nipple discharge [6,7]. Historically, the risk of upgrade to in situ or invasive carcinoma following the diagnosis of a papillary lesion on core needle biopsy (CNB) was as high as 67% [8,9,10]. However, for papillomas without atypia, the upgrade rate was found in a prospective study in 2021 of 117 asymptomatic women to be only 1.7%, suggesting that the increased risk of upgrade with papillary lesions is conferred by the presence atypia [11]. In a retrospective study published that same year for 612 asymptomatic women, the upgrade rate was found to be 3.4%, and factors associated with increased risk of upgrade included discordance between imaging and pathology, age greater than 60 years, lesion size greater than 1 cm and clinical features of nipple discharge or palpable mass [12]. A personalized approach to management is most appropriate with consideration given to each patient’s risk profile, and patients who elect to forego surgery can be closely monitored with serial imaging as most IPAs remain stable during surveillance [13].

### 1.2. Lobular Hyperplasia

In the mid-twentieth century, lobular hyperplasia was considered a precursor lesion and mastectomy was the standard of care [14]. Since then, long-term clinical studies have demonstrated that lobular hyperplasia is better characterized as a marker for elevated lifetime risk of breast cancer development than as a precursor lesion [14]. The upgrade rate for ALH and classic LCIS are low in both prospective and retrospective studies when there is a limited extent of disease and there is concordance between imaging and pathology [15,16,17,18]. In one series when patients diagnosed with cLCIS or ALH were followed with surveillance imaging for up to 5 years, the incidence of progression to malignancy was only 4.8%, and the majority of these carcinomas were detected at separate sites from the initial lesion [17]. Given this modest upgrade rate for cLCIS and ALH, ASBrS guidelines recommend consideration of surveillance in isolated, small-volume, imaging concordant ALH and cLCIS.

### 1.3. Flat Epithelial Atypia and Complex Sclerosing Lesion

Improvement in biopsy techniques and tissue sampling has led to low upgrade rates for FEA and CSL. Previously, upgrade rates for pure FEA and CSLs were up to 9% and 20%, respectively, but most recent studies have reported upgrade rates between 1 and 5% [19]. When the sample size is adequate, there are no additional high-risk lesions identified and the pathology is concordant with imaging, active surveillance can be safely considered for pure FEA and CSL [5].

### 1.4. Shift towards De-Escalation

De-escalation of treatment for high-risk lesions of the breast continues to evolve in response to the identification of features predictive of upgrade. Notably lacking from the studies that have informed contemporary guidelines on the management of these high-risk lesions are cases in which patients are simultaneously diagnosed with in situ or invasive carcinoma. The common use of advanced imaging such as breast magnetic resonance imaging (MRI) during the work-up of carcinoma can lead to the diagnosis of incidental high-risk breast lesions. There is little evidence to guide whether excision should be performed for these lesions when found concurrently with a malignancy. Here, we examine the upgrade rate of these high-risk lesions when diagnosed concurrently with in situ or invasive disease, and identify factors predictive of upgrade or downgrade.

## 2. Materials and Methods

### 2.1. Study Approval

This study was conducted in accordance with U.S. Common Rule. The protocol was approved by the UCSD HRPP under IRB#805822. Study data were collected by one of the authors (J.L.) and managed using Research Electronic Data Capture (REDCap) tools hosted at the University of California San Diego (UCSD) [20,21]. REDCap is a secure, web-based software platform designed to support data capture for research studies.

### 2.2. Data Extraction

Utilizing a prospectively maintained institutional database, a retrospective review of electronic medical records was performed for females who were newly diagnosed with breast carcinoma with simultaneous diagnosis of separate-site high-risk breast lesion between 1 January 2006 and 31 December 2021. High-risk breast lesions included IPA, cLCIS, ALH, FEA and CSL. Inclusion criteria were females aged 18 and older treated with partial mastectomy and simultaneous separate excision of high-risk breast lesion. Exclusion criteria included patients who underwent non-partial mastectomies and those who had missing data on the sizes of the excised malignancy and high-risk lesion. Data were curated on patient demographics, family history of malignancy, genetic testing, clinicopathologic tumor characteristics, receipt of systemic and radiation therapies, type of imaging modality and imaging characteristics of the benign breast lesions. 

### 2.3. Statistical Analysis

Continuous variables were transformed into categorical variables with an attempt to categorize the full cohort into three or four groups of comparable sizes for each variable. Regression models were developed to identify preoperative risk factors associated with an upgrade to malignancy (invasive carcinoma or ductal carcinoma in situ [DCIS]) or a downgrade to no evidence of malignancy or high-risk lesion. Univariate models were developed for each variable collected, and all variables that were significant at *p* less than 0.05 were incorporated into a multivariable model. MRI descriptors (irregular, linear, heterogeneous, oval, circumscribed, regional, clumped, linear, segmental, spiculated, round, focal) were assessed separately for the subset of patients who underwent MRI using chi-square testing comparing patients who had an upgrade to those who did not have lesion upgrade. All statistical analysis was performed in R (version 4.0.3, R Foundation for Statistical Computing, Vienna, Austria) using RStudio (Version 1.1.463) and packages “tidyverse” (Version 1.3.0) and “nnet” (Version 4.2.3) [22,23].

## 3. Results

### 3.1. Cohort Description

A total of 65 patients meeting the inclusion criteria were identified. The median age of patients was 60 years (interquartile range [IQR] 50–66 years), median body mass index was 26 (IQR 23–31) and most patients were either Caucasian (*n* = 31, 47.7%) or Hispanic (*n* = 14, 21.5%). Among them, 5 (7.7%) patients had an upgrade of their lesion to carcinoma (4 upgraded to DCIS and 1 upgraded to low grade invasive ductal carcinoma), 18 (27.7%) had no evidence of malignancy or high-risk lesion on final pathology and 42 (64.6%) had persistence of their high-risk lesion on final pathology (Table 1, Figure 1). There were 22 patients who underwent genetic testing, of whom only two had mutations identified, both of which were CHEK2 mutations. One of these two patients had an upgrade of her high-risk lesion site to invasive disease, and the other had persistence of her high-risk lesion. Five patients had a past personal history of non-oncologic breast surgery, and all five of them had persistence of their high-risk lesions on final pathology. 

The concurrently identified malignancy was an invasive ductal carcinoma in the majority (*n* = 42, 64.6%) of patients, with most having estrogen receptor (ER) positive (*n* = 50, 76.9%) disease, and most having HER2 negative (*n* = 33, 50.8%) disease. Most patients had T1 (*n* = 28, 43.1%) or T2 (*n* = 23, 35.4%) disease, and most had grade II disease (*n* = 35, 53.8%). The malignancy was identified by a mammogram in 44 (67.7%) cases, was self-detected by palpation by the patient in 12 (18.5%) cases, was identified on ultrasound in 7 (10.8%) cases and by MRI in 2 (3.1%) cases. In contrast, the high-risk lesion was detected by MRI in 45 (69.2%) cases, by mammogram in 10 (15.4%) cases and by ultrasound in 10 (15.4%) cases. For 44 (67.7%) patients, the high-risk lesion was in the breast contralateral to the malignancy. Most high-risk lesions identified were IPA (*n* = 38, 58.5%), and the median of the greatest dimension of the lesion as measured by any preoperative imaging modality was 1 cm (IQR 0.6–2.2 cm). Most high-risk lesions were in the upper-outer (*n* = 26, 40%) or lower-outer (*n* = 16, 24.6%) quadrants of the breast.

Most patients (*n* = 47, 72.3%) received neither neoadjuvant chemotherapy nor neoadjuvant endocrine therapy (*n* = 62, 95.4%). Carcinomas were excised to negative pathologic margins (no tumor on ink). The mean of the greatest dimension of the lumpectomy specimen for the high-risk lesion was 5.07 cm and the mean of the greatest dimension of the lumpectomy specimen for the malignancy was 7.00 cm (*p* = 0.003). The mean of the greatest dimension of the final pathology of the high-risk lesion was 0.86 cm and the mean of the greatest dimension of the final pathology of the malignancy was 1.50 cm (*p* < 0.001) (Figure 2).

### 3.2. MRI Features of High-Risk Lesions

Among the 46 (70.8%) patients who underwent MRI, 21 (45.7%) were able to be assessed for volumetric analysis of the high-risk lesion with median estimated volume of 0.19 cm^3^ (IQR 0.1–0.4 cm^3^) and the median of the greatest dimension of the high-risk lesions as measured on MRI was 1.05 cm (IQR 0.6–2.2 cm). Half of the lesions had a mass-like appearance on MRI (*n* = 23, 50%) and most of the remaining lesions had the appearance of non-mass enhancement (*n* = 19, 41.3%). Patients with oval lesions (*p* = 0.01) and lesions that were circumscribed (*p* = 0.02) were more likely to upgrade than to have a persistent high-risk lesion or no evidence of disease on final pathology.

### 3.3. Variables Utilized in Univariate Regression Models

The following variables were tested in the univariate models for consideration of incorporation into the multivariate model: age, BMI, race/ethnicity, genetic testing results, family history of breast cancer, personal history of breast cancer, personal breast surgical history, history of hormone replacement therapy, method of imaging detection of the malignancy, laterality of the malignancy, quadrant of the malignancy, type of malignancy (e.g., IDC, ILC, DCIS), grade of malignancy, ER status of malignancy, progesterone receptor (PR) status of malignancy, HER2 status of malignancy, T stage, N stage and M stage of malignancy, as well as overall clinical TNM stage, mode of imaging on which the high-risk lesion was identified, type of high-risk lesion identified (divided into three subgroups: papilloma, radial scar/complex sclerosing lesion and lobular carcinoma in situ [LCIS]/atypical lobular hyperplasia [ALH]/flat epithelial atypia [FEA]), quadrant of the high-risk lesion, whether the lesion was ipsilateral and within 5 cm of the malignancy vs. contralateral or greater than 5 cm from the malignancy, American Society of Anesthesiologists classification, the type of biopsy performed for the high-risk lesion preoperatively, the gauge of the needle used for the biopsy, the estimated preoperative size of the lesion on imaging, the estimated size of the high-risk lesion after receipt of neoadjuvant therapy when administered, whether neoadjuvant endocrine therapy or neoadjuvant chemotherapy was administered and the duration between the time of the biopsy diagnosing cancer and the time of the biopsy diagnosing the high-risk lesion. Margin status of the high-risk lesion was not assessed because it was not documented in pathology reports, as the margin status of high-risk lesions has been shown to not influence outcomes nor management [24,25,26,27,28]. Lifestyle factors such as diet, smoking and alcohol use are also not reliably recorded and so were not curated and assessed.

### 3.4. Logistic Regression Model for Lesion Upgrade or Downgrade

An ordinal logistic regression model was initially developed, but the resultant multivariate model was found to violate the assumption of proportional odds, so a multinomial logistic regression model was developed instead (Table 2). In the univariate analysis, the presence of the high-risk lesion in the upper inner quadrant was found to be associated with downgrade of disease (odds ratio [OR] = 11.9, *p* = 0.009), and the presence of the high-risk lesion in the same breast as the malignancy and within a 5 cm distance from it was associated with increased risk of upgrade (OR = 12.0, *p* = 0.04). In the multivariate model, location of the high-risk lesion in the upper inner quadrant was still associated with downgrade (OR = 12.2, *p* = 0.009) and presence of the high-risk lesion within 5 cm of the malignancy in the ipsilateral breast was still associated with upgrade to invasive disease (OR = 12.7, *p* = 0.03). Among patients who underwent preoperative MRI, the median high-risk lesion size was 1.05 cm (IQR 0.6–2.2) and on the final pathology the median high-risk lesion size was 0.85 cm (IQR 0–1.2 cm), and the differences in the means was significant (*p* = 0.003).

## 4. Discussion

Here, we demonstrate that for high-risk lesions diagnosed concurrently with breast malignancy, the risk of the high-risk lesion harboring carcinoma depends on its distance and laterality from the malignancy and the quadrant of the breast in which it is located. This is important because while the management of isolated high-risk breast lesions continues to move towards de-escalation, the proper approach to management of these incidentally identified high-risk breast lesions in the context of concurrently diagnosed carcinoma has not been well delineated.

One approach to the stratification of risk conferred by these high-risk lesions has been the study of imaging features on MRI, as many of these lesions are diagnosed by this modality. Some authors have reported modestly increased risk of upgrade with ALH, LCIS, CSL and IPA lesions that are large in size, have clumped non-mass enhancement (NME) with mixed kinetics or NME with regional/segmental distribution [29,30,31]. Other authors have not found any imaging features predictive of upgrade. In one series of 61 MRI-detected, CNB confirmed high-risk lesions of ALH, LCIS and radial scar, Strigel et al. found that none of the MRI features studied were predictive of upgrade [32]. In a meta-analysis of upgrade rates for a wider range of high-risk lesions including atypical ductal hyperplasia (ADH), ALH, LCIS, IPA, radial scar, FEA, mucocele-like lesions and phyllodes tumors, Heller et al. were unable to identify any MRI morphologic or kinetic qualities of high-risk lesions that were predictive of upgrade to carcinoma [33]. In agreement with this meta-analysis, we were unable to identify any imaging features predictive of upgrade including lesion size, mass versus NME, shape (oval versus not), irregularity or heterogeneity. 

While we were unable to identify any MRI features predictive of upgrade to malignancy in this study of patients with high-risk lesions diagnosed concurrently with carcinoma, we did identify other disease features that were predictive. The presence of the high-risk lesion in the same breast as the carcinoma and within a 5 cm distance from it was associated with an increased risk of upgrade. Most lesions that upgraded to malignancy on final pathology were IPA, although the proportion of upgraded patients who had a high-risk lesion of IPA (60%) was comparable to the proportion of patients with a high-risk lesion of IPA in the full patient cohort (58.5%). Han et al. reported that for patients diagnosed with IPAs, the presence of a concurrent contralateral breast cancer was associated with an increased rate of upgrade of the IPAs to carcinoma, though curiously their series either did not contain any patients with concurrently diagnosed ipsilateral carcinomas, or else they did not include this feature in their analysis [34]. In contrast, later studies by Wang et al. demonstrated that concurrent contralateral or ipsilateral breast cancer distinct from the IPA was an independent risk indicator for upgrade to carcinoma [29]. When considering upgrade rates reported in other contemporary series, our study is consistent with this finding of increased upgrade rate when an IPA without atypia is identified in a patient who has a concurrently identified malignancy. Our regression analysis also identified presence of the lesion in the upper inner quadrant of the breast as a feature associated with no evidence of disease on final pathology, though given the relatively small proportion of patients with disease in that quadrant (13.8%), this finding in the regression analysis may have been secondary to the small sample size. 

An additional consideration in the approach to management for these high-risk breast lesions is the relative risk of upgrade in relation to the anticipated benefit of high-risk lesion excision. MRI tends to overestimate disease inclusive of both invasive and benign breast pathology which was reflected in our data where the high-risk lesions appeared larger on the MRI compared to surgical pathology leading to more tissue excised [35]. The median size of the greatest dimension of breast carcinoma on final pathology in this patient cohort was comparable to that of the median size on the pathology of the high-risk lesion. The median size of the greatest dimension of the excised specimen from the high-risk lesion site was more than half that of the cancer specimen (4.3 cm vs. 7.1 cm), so patients underwent a sizeable additional resection for a lesion that was upgraded to carcinoma in only 7.7% of cases. Not only does this affect total volume excised and subsequent cosmesis, but also operative duration and risk of post-operative complications such as hematoma and infection, since the separate-site surgery increases the number of sites at which such complications can occur.

The primary limitation of this study is the small sample size, both with respect to the overall cohort and when considering each specific type of high-risk lesion. This limits the generalizability of our findings for each type of lesion since several of them were grouped together and the lesion type was not predictive of lesion upgrade. Other limitations include the retrospective, non-randomized, single-institution nature of the study and the inconsistent availability of data on the sizes of the high-risk lesions on pathology and on the MRI. The greatest strength of this study is the reliability of the data that were collected, as all variables were clearly defined prior to initiation of data curation, and data collection was performed by a single practicing breast surgeon.

## 5. Conclusions

The existing medical literature on the management of concurrently diagnosed high-risk breast lesions diagnosed concurrently with a separate site breast carcinoma is sparse. In our single-institution cohort study, we found that lesions within a 5 cm distance of an ipsilateral breast carcinoma should be considered for excision given the high upgrade rate of 25%. For other patients who are considered on comprehensive assessment to be of lower risk for upgrade of disease, omission of surgery can be considered.

## Figures and Tables

**Figure 1 cancers-16-02268-f001:**
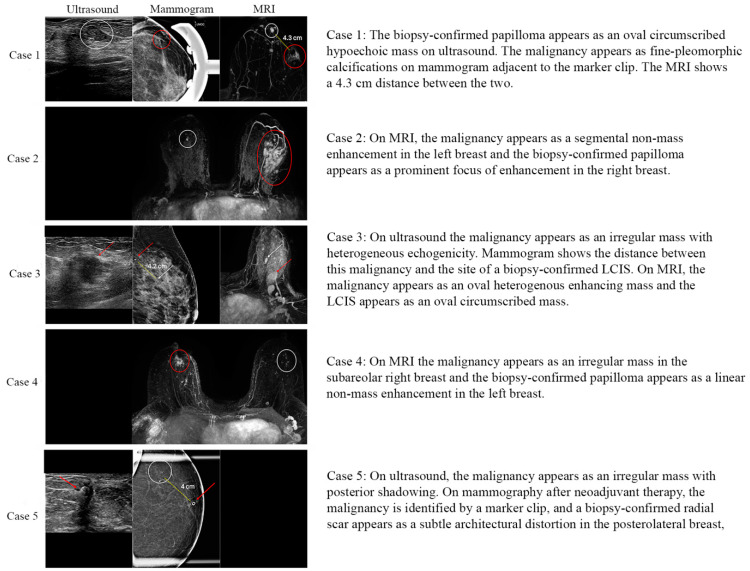
Imaging characteristics of the five upgraded high-risk lesions to in situ or invasive carcinoma. White circles and arrows mark high-risk lesions, red circles and arrows mark concurrent malignancies and yellow dashed lines indicate the distance between them.

**Figure 2 cancers-16-02268-f002:**
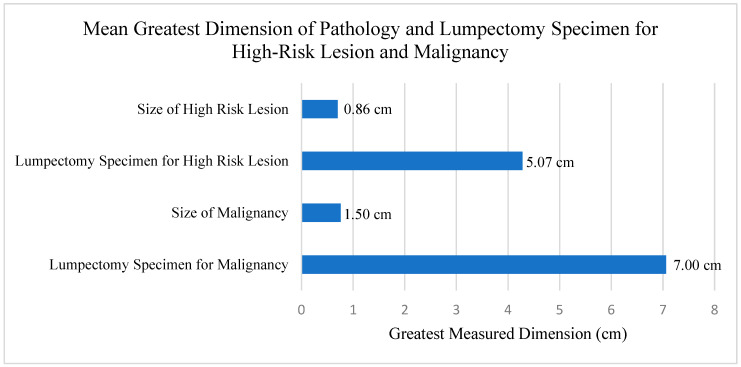
Mean size of high-risk lesion and malignancy, and associated lumpectomy sizes for high-risk lesion and malignancy. Patients who had a downgrade of their lesions as well as patients who had no evidence of malignancy were included in calculation of means.

**Table 1 cancers-16-02268-t001:** Characterization of patient cohort. Full cohort characterized, as well as sub-groups according to whether patients had no evidence of disease identified on the surgical specimen for their high-risk lesion (“Downgrade”), persistence of the high-risk lesion without any evidence of invasive disease (“Persistent High-Risk Lesion”) or had evidence of invasive disease on the surgical specimen for their high-risk lesion (“Upgrade”).

Variable		Full (*n* = 65)	Upgrade (*n* = 5)	Downgrade (*n* = 18)	Persistent High-Risk Lesion (*n* = 42)
Age (years)	≤50	17 (26.2%)	1 (20%)	7 (38.9%)	9 (21.4%)
	50–60	16 (24.6%)	3 (60%)	3 (16.7%)	10 (23.8%)
	60–65	15 (23.1%)	1 (20%)	4 (22.2%)	10 (23.8%)
	>65	17 (26.2%)	0 (0%)	4 (22.2%)	13 (31%)
Race	Caucasian	31 (47.7%)	2 (40%)	14 (77.8%)	15 (35.7%)
	African American	3 (4.6%)	0 (0%)	0 (0%)	3 (7.1%)
	Asian	10 (15.4%)	2 (40%)	1 (5.6%)	7 (16.7%)
	Hispanic	14 (21.5%)	1 (20%)	2 (11.1%)	11 (26.2%)
	Other	7 (10.8%)	0 (0%)	1 (5.6%)	6 (14.3%)
Modality of carcinoma detection	Self-palpated	12 (18.5%)	1 (20%)	5 (27.8%)	6 (14.3%)
	Mammogram	44 (67.7%)	4 (80%)	9 (50%)	31 (73.8%)
	Ultrasound	7 (10.8%)	0 (0%)	3 (16.7%)	4 (9.5%)
	MRI	2 (3.1%)	0 (0%)	1 (5.6%)	1 (2.4%)
Imaging modality of high-risk lesion detection	MRI	46 (70.8%)	4 (80%)	13 (72.2%)	29 (69%)
	Mammography	9 (13.8%)	1 (20%)	1 (5.6%)	7 (16.7%)
	Ultrasound	10 (15.4%)	0 (0%)	4 (22.2%)	6 (14.3%)
Type of high-risk lesion	Papilloma	38 (58.5%)	3 (60%)	8 (44.4%)	27 (64.3%)
	CSL	17 (26.2%)	1 (20%)	7 (38.9%)	9 (21.4%)
	LCIS, ALH or FEA	10 (15.4%)	1 (20%)	3 (16.7%)	6 (14.3%)
Size of high-risk lesion on preoperative imaging (cm)	<0.75	25 (38.5%)	0 (0%)	9 (50%)	16 (38.1%)
	0.75–1.5	20 (30.8%)	2 (40%)	5 (27.8%)	13 (31%)
	>1.5	20 (30.8%)	3 (60%)	4 (22.2%)	13 (31%)
Quadrant location of high-risk lesion	Upper outer	26 (40%)	4 (80%)	5 (27.8%)	17 (40.5%)
	Lower outer	16 (24.6%)	1 (20%)	5 (27.8%)	10 (23.8%)
	Central, subareolar or lower inner	14 (21.5%)	0 (0%)	1 (5.6%)	13 (31%)
	Upper inner	9 (13.8%)	0 (0%)	7 (38.9%)	2 (4.8%)
Location of high-risk lesion relative to carcinoma	Contralateral or Ipsilateral >5 cm	53 (81.5%)	2 (40%)	16 (88.9%)	35 (83.3%)
	Ipsilateral, ≤5 cm	12 (18.5%)	3 (60%)	2 (11.1%)	7 (16.7%)
Carcinoma Type	Invasive ductal carcinoma	42 (64.6%)	3 (60%)	14 (77.8%)	25 (59.5%)
	Ductal carcinoma in situ	11 (16.9%)	2 (40%)	1 (5.6%)	8 (19%)
	Invasive lobular carcinoma	5 (7.7%)	0 (0%)	2 (11.1%)	3 (7.1%)
	Mixed invasive ductal and lobular	3 (4.6%)	0 (0%)	0 (0%)	3 (7.1%)
	Other	4 (6.2%)	0 (0%)	1 (5.6%)	3 (7.1%)
Estrogen receptor status	Positive	50 (76.9%)	4 (80%)	13 (72.2%)	33 (78.6%)
	Negative	15 (23.1%)	1 (20%)	5 (27.8%)	9 (21.4%)
Progesterone receptor status	Positive	44 (67.7%)	4 (80%)	10 (55.6%)	30 (71.4%)
	Negative	21 (32.3%)	1 (20%)	8 (44.4%)	12 (28.6%)
HER2 IHC status	Negative	33 (50.8%)	2 (40%)	11 (61.1%)	20 (47.6%)
	Borderline	12 (18.5%)	0 (0%)	5 (27.8%)	7 (16.7%)
	Positive	7 (10.8%)	0 (0%)	1 (5.6%)	6 (14.3%)
	NA	13 (20%)	3 (60%)	1 (5.6%)	9 (21.4%)
Clinical stage	Stage 0	11 (16.9%)	2 (40%)	1 (5.6%)	8 (19%)
	Stage 1	24 (36.9%)	0 (0%)	7 (38.9%)	17 (40.5%)
	Stage 2	25 (38.5%)	3 (60%)	8 (44.4%)	14 (33.3%)
	Stage 3	4 (6.2%)	0 (0%)	2 (11.1%)	2 (4.8%)
	Stage 4	1 (1.5%)	0 (0%)	0 (0%)	1 (2.4%)
Pathologic stage	No residual invasive disease (ypT0N0)	5 (7.7%)	0 (0%)	2 (11.1%)	3 (7.1%)
	ypTisN0	2 (3.1%)	1 (20%)	1 (5.6%)	0 (0%)
	ypT+ or ypN+	13 (20%)	2 (40%)	5 (27.8%)	6 (14.3%)
	Stage 0	9 (13.8%)	1 (20%)	1 (5.6%)	7 (16.7%)
	Stage 1	29 (44.6%)	1 (20%)	9 (50%)	19 (45.2%)
	Stage 2	4 (6.2%)	0 (0%)	0 (0%)	4 (9.5%)
	Stage 3	2 (3.1%)	0 (0%)	0 (0%)	2 (4.8%)
	Stage 4	1 (1.5%)	0 (0%)	0 (0%)	1 (2.4%)
Neoadjuvant chemotherapy received	No	47 (72.3%)	3 (60%)	11 (61.1%)	33 (78.6%)
	Yes	18 (27.7%)	2 (40%)	7 (38.9%)	9 (21.4%)
Neoadjuvant endocrine therapy received	No	62 (95.4%)	4 (80%)	17 (94.4%)	41 (97.6%)
	Yes	3 (4.6%)	1 (20%)	1 (5.6%)	1 (2.4%)

**Table 2 cancers-16-02268-t002:** Multivariate multinomial logistic regression model. No disease indicates that neither carcinoma nor a high-risk lesion was identified on final pathology at the site of previously biopsy-confirmed high-risk lesion. Upgrade to carcinoma designates the finding of either in situ or invasive carcinoma at the site of previously biopsy-confirmed high-risk lesion. Persistence of the high-risk lesion on final pathology was the reference outcome.

	No Disease		Upgrade to Carcinoma	
	Odds Ratio (95% CI)	*p*-Value	Odds Ratio (95% CI)	*p*-Value
Distance from malignancy				
Contralateral or ipsilateral, >5 cm	Ref	-	Ref	-
Ipsilateral, ≤5 cm	0.65 (0.10–4.46)	0.66	12.7 (1.32–121.9)	0.03
Quadrant of high-risk lesion				
Upper outer	Ref	-	Ref	-
Upper inner	12.2 (1.88–78.8)	0.009	N/A	-
Lower outer	1.79 (0.41–7.94)	0.44	0.23 (0.02–3.19)	0.27
Central, subareolar or lower inner	0.27 (0.03–2.67)	0.27	N/A	-

CI ≡ Confidence interval; N/A ≡ not applicable, no events of interest observed; Ref ≡ defined reference group.

## Data Availability

The data that were curated for this manuscript are unavailable due to privacy concerns.

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
