# Peer review of "Distance of Biopsy-Confirmed High-Risk Breast Lesion from Concurrently Identified Breast Malignancy Associated with Risk of Carcinoma at the High-Risk Lesion Site"

_cancers, 2024, doi:10.3390/cancers16122268_

Round 1

Reviewer 1 Report

Comments and Suggestions for Authors

Distance of Biopsy-Confirmed High-Risk Breast Lesion from Concurrently Identified Breast Malignancy Associated with Risk of Carcinoma at the High-Risk Lesion Site by Le demonstrate Incidental intraductal papilloma without atypia (IPA), lobular hyperplasia (LCIS 13 or ALH), flat epithelial atypia (FEA) and complex sclerosing lesion (CSL) are high-risk lesions of the 14 breasts. These lesions are not routinely excised when diagnosed in isolation because they have low rates of upgrade to invasive cancer. When identified concurrently with invasive breast cancer, however, the upgrade rate is not well-characterized. Authors sought to both characterize the upgrade rate for these lesions when diagnosed concurrently with invasive breast cancer, and to identify features of the high-risk lesions predictive of upgrade to invasive cancer. In our cohort of 65 patients who were concurrently diagnosed with a high-risk breast lesion and an invasive cancer, 5 patients (7.7%) had upgrade of their high-risk lesion to carcinoma. The rate of upgrade was higher for high-risk lesions that were ipsilateral to malignancy and within 5 cm of it, and lower for other lesions.  

Comments to the Editor and Author.        

Basically, the number of figures is limited.

  1. Fig 1. Compares Ultrasound, Mammogram and MRI for different cases.  The authors could show more cases for each category will be great.
  2. Table 1. Show more ‘N’ will increase the quality.
  3. Fig 2. Please include p values with more ‘N’ numbers.
  1. Does the Fig 1 correlate with any clinical markers?

Author Response

Manuscript: Distance of Biopsy-Confirmed High-Risk Breast Lesion from Concurrently Identified Breast Malignancy Associated with Risk of Carcinoma at the High-Risk Lesion Site

Reviewer Comments (bolded) and Responses:

Reviewer #1 - Basically, the number of figures is limited.

  1. Fig 1. Compares Ultrasound, Mammogram and MRI for different cases.  The authors could show more cases for each category will be great.
  2. Table 1. Show more ‘N’ will increase the quality.
  3. Fig 2. Please include p values with more ‘N’ numbers.
  4. Does the Fig 1 correlate with any clinical markers?

We thank reviewer #1 for their thoughtful feedback on our manuscript.

  1. We appreciate reviewer #1’s comment regarding the increased number of cases for each category. We only included patients who had upgrade of their high-risk lesions because the number of cases is small enough to allow for depiction of each of these lesions, and also because of the wide variability in the imaging findings of the non-upgraded patients. The inclusion of additional images from other cases would neither be representative of the non-upgraded patients, nor would it meaningfully allow for contrast with the upgraded patients. We did not find imaging features predictive of upgrade so there are no groupings of images that would illustrate any concepts from our analysis. If after this explanation reviewer #1 still feels that the addition of more cases/images would improve the quality of the manuscript, we are happy to do so.

  1. We appreciate reviewer #1’s comment regarding the number of patients assessed for this study. While we agree that a larger number of patients would improve the study, the only way to increase the number further would be to extend the start date of the study prior to 2006. This would result in a higher proportion of patients with paper charts rather than electronic medical records, leading to increased rates of incomplete data, as well as an increased risk of data being available but not identified due to factors like illegible handwriting, unorganized records within the paper charts, failure to replace records from a paper chart, and so on. Furthermore, the further back in time the study is extended, the less likely that these high-risk lesions, many of which are mammographically occult, would have been identified, so the proportion of patients otherwise meeting criteria would become smaller. However, completing additional review would take longer than 10 days to complete, and given the low upgrade rate coupled with the anticipated low rate of patients identified further back, we suspect it is unlikely that this will change the results of our analysis.

  1. We appreciate reviewer #1’s comment regarding the lack of comparisons of sizes in our Figure 2. The N is already fixed at 65. We did not feel that a demonstration between differences in the malignancy and high-risk lesion sizes or the lumpectomy sizes of each would have clinical importance but we have now updated the figure and included the following which switches from medians to means to allow for interpretation of these comparisons (Results 3.1, lines 168-173):
    “The mean of the greatest dimension of the lumpectomy specimen for the high-risk lesion was 5.07 cm and the mean of the greatest dimension of the lumpectomy specimen for the malignancy was 7.00 cm (p=0.003). The mean of the greatest dimension of the final pathology of the high-risk lesion was 0.86 cm and the mean of the greatest dimension of the final pathology of the malignancy was 1.50 cm (p<0.001).”

  1. We appreciate reviewer #1’s question regarding Figure 1. The features described in Figure 1 were the imaging features for the patients who had high-risk lesions that upgraded. They were not associated with any other clinical markers.

Reviewer 2 Report

Comments and Suggestions for Authors

1. The study should include data on the surgical margin status because it is important for determining if all of the high-risk lesion was removed. Positive margins might require additional surgery. Why did the study not include data on the surgical margin status of high-risk lesions? Knowing if the margins were clear or positive can affect the decision for further surgery.

2. The study did not specify the imaging characteristics of non-mass enhancements (NME). Including specific MRI features such as the shape or margin of non-mass enhancements can help predict which high-risk lesions are more likely to upgrade to cancer. Can you explain if there were any specific MRI features like shape or margin that predicted upgrade? Looks like the study has inconsistent definitions. The study does not clearly define what constitutes an "upgrade" in all cases, which can lead to inconsistent interpretation of results.

3. The study did not provide details on the histological subtype of upgraded carcinomas. Were there any specific histological subtypes more common in upgraded lesions? The study should include details on the histological subtype of upgraded carcinomas because certain subtypes may have different prognostic and treatment implications.

4. Why did the study not discuss the potential impact of hormonal receptor status (ER, PR) on the upgrade of high-risk lesions to carcinoma? Discussing the impact of hormonal receptor status is important because ER and PR positive lesions may have different risks and treatment options compared to hormone receptor-negative lesions.

5. Hormone Replacement Therapy. The study does not mention if patients were on hormone replacement therapy (HRT). HRT can affect breast cancer risk and progression.

6. Lifestyle Factors. There is no information on lifestyle factors such as diet, smoking, or alcohol use, which can influence breast cancer risk.

7. The study did not address potential confounders. The study should include an analysis of potential confounders such as family history and genetic mutations because these factors can influence the risk of upgrading to cancer. How did you control for other risk factors, like family history or genetic mutations, that might influence the upgrade rate? Treatment Consistency: Variations in treatment protocols over the 15-year study period are not addressed. Consistent treatment approaches are important for reliable results. Age: Younger and older patients might have different risks of upgrade. Genetic Mutations: Specific genetic mutations like BRCA1 or BRCA2 can significantly alter cancer risk. Family History: A family history of breast cancer can be a strong predictor of upgrade. Hormone Receptor Status: ER/PR positive or negative status can influence the likelihood of progression to carcinoma. Include interaction terms in the regression models to explore if the effect of one variable depends on the level of another variable could be interesting.

8. The study included genetic testing results but did not specify the types of mutations found. Were there specific genetic mutations associated with a higher risk of lesion upgrade? Specifying the types of genetic mutations found and their association with lesion upgrade can provide more insight into which patients are at higher risk.

9. What kind of follow-up care was provided to patients with high-risk lesions that did not undergo excision? The study does not specify the follow-up duration for patients. Long-term follow-up is important to understand the true upgrade risk of high-risk lesions. Including information on patient follow-up care is important to understand how these patients were monitored over time and to ensure any progression was detected early.

10. The study did not consider the cost-effectiveness of excising high-risk lesions. Did you evaluate if routine excision of these lesions is cost-effective compared to surveillance? Evaluating the cost-effectiveness of excising high-risk lesions versus surveillance is important because it can impact healthcare policies and recommendations.

11. The study did not perform a power analysis. Did the study have sufficient power to detect differences in upgrade rates between different high-risk lesions? Performing a power analysis is important to ensure that the study had enough participants to detect meaningful differences in upgrade rates, providing confidence in the results.

12. The study did not discuss the impact of the interval between the diagnosis of high-risk lesions and invasive cancer. How did the time interval affect the upgrade rates? Discussing the impact of the time interval between diagnoses is important because it can influence the progression of high-risk lesions to cancer. Longer intervals might show different upgrade rates compared to shorter ones.

 These questions and critiques should be added to the discussion section of the article to address potential limitations and provide a more comprehensive understanding of the findings. If there are specific sections where the data is missing, they should be added to ensure the study is thorough and well-rounded.

Here also are the main limitations of this retrospective cohort study that must be cited in the discussion. These limitations mean that the study's findings should be interpreted carefully and may not be broadly applicable. Please review this in the text:

a) The study included only 65 patients. This is a small number, which makes it hard to apply the findings to a larger population.

b) The ethnic and racial diversity. The study did not thoroughly analyze the impact of ethnic and racial diversity on the upgrade rates. Different populations may have different risks.

c) The study depends on past records, which might have missing or incorrect information.

d) Researchers cannot control how data was originally collected. This can affect the consistency and quality of the information.

e) Looking back at past data means researchers cannot influence the study design or data collection, which might limit the study's reliability.

f) There could be selection bias because the study only includes patients who were treated at one specific center and met certain criteria. The study was conducted in a single center. This means the sample may not represent the broader population, limiting the generalizability of the results.

g) Only patients who underwent surgery were included. This could introduce selection bias because patients who did not have surgery might have different outcomes.

h) Inconsistent Definitions. The study does not clearly define what constitutes an "upgrade" in all cases, which can lead to inconsistent interpretation of results.

i) The results might not apply to all patients with similar conditions because the study was done in one hospital with specific patient demographics.

j) Lack of randomization. As a retrospective study, there is no randomization. This can introduce various biases and affect the study’s validity.

k) Patient Preferences: The study does not consider patient preferences or the psychological impact of knowing they have a high-risk lesion. This can influence decisions on treatment and follow-up.

l) Treatment Consistency: Variations in treatment protocols over the 15-year study period are not addressed. Consistent treatment approaches are important for reliable results.

Author Response

Manuscript: Distance of Biopsy-Confirmed High-Risk Breast Lesion from Concurrently Identified Breast Malignancy Associated with Risk of Carcinoma at the High-Risk Lesion Site

Reviewer Comments (bolded) and Responses:

We thank reviewer #2 for their thorough review of our manuscript and their feedback.

  1. The study should include data on the surgical margin status because it is important for determining if all of the high-risk lesion was removed. Positive margins might require additional surgery. Why did the study not include data on the surgical margin status of high-risk lesions? Knowing if the margins were clear or positive can affect the decision for further surgery.

We appreciate this question regarding surgical margin status by reviewer #2. All carcinomas were excised to negative pathologic margins (no tumor on ink). The margins of high-risk lesions are not recorded at our institution since margin status has not been found to influence patient outcomes, and so does not influence management. We included the following in the results section to clarify this (Results 3.1, lines 171-172; Results 3.3, lines 210-213):

“Carcinomas were excised to negative pathologic margins (no tumor on ink).”

“Margin status of the high-risk lesion was not assessed because it was not documented in pathology reports, as margin status of high-risk lesions has been shown to not influence outcomes nor  management.24-28”

  1. The study did not specify the imaging characteristics of non-mass enhancements (NME). Including specific MRI features such as the shape or margin of non-mass enhancements can help predict which high-risk lesions are more likely to upgrade to cancer. Can you explain if there were any specific MRI features like shape or margin that predicted upgrade? Looks like the study has inconsistent definitions. The study does not clearly define what constitutes an "upgrade" in all cases, which can lead to inconsistent interpretation of results.

We thank reviewer #2 for their question regarding the imaging characteristics and the definition of upgrade.

Regarding the former, since MRI was only performed on 46/65 patients, and 1/5 patients who had an upgrade of their lesion did not have an MRI, we assessed MRI descriptors only as a subset analysis by Chi-square testing. We have included the following into Methods (Methods 2.3, lines 123-127):

“MRI descriptors were assessed separately for the subset of patients who underwent MRI using Chi-square testing comparing patients who had upgrade to those who did not have lesion upgrade.”

And the following into Results (Results 3.2 lines 188-190):

“Patients with oval lesions (p=0.01) and lesions that were circumscribed (p=0.02) were more likely to upgrade than to have a persistent high-risk lesion or no evidence of disease on final pathology.”

Regarding the latter, the following was added to the methods (Methods 2.3, lines 119-121)

“Regression models were developed to identify preoperative risk factors associated with upgrade to malignancy (invasive carcinoma or ductal carcinoma in situ [DCIS]) or downgrade to no evidence of malignancy or high-risk lesion.”

Also, the following sentence was edited in the simple summary to (lines 17-19)

“We sought to both characterize the upgrade rate for these lesions when diagnosed concurrently with invasive breast cancer, and to identify features of the high-risk lesions predictive of upgrade to cancer.”

  1. The study did not provide details on the histological subtype of upgraded carcinomas. Were there any specific histological subtypes more common in upgraded lesions? The study should include details on the histological subtype of upgraded carcinomas because certain subtypes may have different prognostic and treatment implications.

We thank reviewer #2 for their question regarding the subtype of upgraded carcinomas.

We edited the following sentence in the results to add “low grade” and “ductal” before mention of the invasive upgrade (Results 3.1, lines 135-137):

“Among them, 5 (7.7%) patients had an upgrade of their lesion to carcinoma (4 upgraded to DCIS and 1 upgraded to low grade invasive ductal carcinoma)”.

  1. Why did the study not discuss the potential impact of hormonal receptor status (ER, PR) on the upgrade of high-risk lesions to carcinoma? Discussing the impact of hormonal receptor status is important because ER and PR positive lesions may have different risks and treatment options compared to hormone receptor-negative lesions.

We thank reviewer #2 for their question regarding the assessment of hormone receptor status on the upgrade rate. As discussed in the manuscript, we did assess for hormone status. It was listed under section 3.3 “Variables Utilized in Univariate Regression Models section.” It was not significant in the univariate model and so was not used in the multivariate model, as described in the methods: “Univariate models were developed for each variable collected, and all variables that were significant at p less than 0.05 were incorporated into a multivariable model.“ (lines 122-124)

  1. Hormone Replacement Therapy. The study does not mention if patients were on hormone replacement therapy (HRT). HRT can affect breast cancer risk and progression.

We thank reviewer #2 for their question regarding hormone replacement therapy. To address this, we have now curated data on patients who received hormone replacement therapy. It was not significant in the univariate model and so did not pass to the multivariate model. (Results 3.3, line 195)

  1. Lifestyle Factors. There is no information on lifestyle factors such as diet, smoking, or alcohol use, which can influence breast cancer risk.

We thank reviewer #2 for their question regarding lifestyle factors. Such factors are not reliably collected and so retrospectively curating this data would represent a highly inaccurate estimate. Furthermore, the fact that all of these patients have concurrent carcinoma represents a greater risk for potential upgrade of high-risk lesion than diet/smoking/alcohol use. We have added the following line to the results section to explain this (Results 3.3, lines 213-214):

“Lifestyle factors such as diet, smoking, and alcohol use are also not reliably recorded and so were not curated and assessed.”

  1. The study did not address potential confounders. The study should include an analysis of potential confounders such as family history and genetic mutations because these factors can influence the risk of upgrading to cancer. How did you control for other risk factors, like family history or genetic mutations, that might influence the upgrade rate? Treatment Consistency: Variations in treatment protocols over the 15-year study period are not addressed. Consistent treatment approaches are important for reliable results. Age: Younger and older patients might have different risks of upgrade. Genetic Mutations: Specific genetic mutations like BRCA1 or BRCA2 can significantly alter cancer risk. Family History: A family history of breast cancer can be a strong predictor of upgrade. Hormone Receptor Status: ER/PR positive or negative status can influence the likelihood of progression to carcinoma. Include interaction terms in the regression models to explore if the effect of one variable depends on the level of another variable could be interesting.

We thank reviewer #2 for their question regarding confounders.

Regarding the suggestion that we consider family history, genetic mutations, age, and hormone receptor status, these factors were all included in our study. As described in the variables assessed (lines 122-124), these variables were each assessed in a univariate model, and variables that were significant were included in a multivariate model. Both forward regression selection and backward elimination are frequently used methods to develop multivariate models in retrospective studies. Both produced the same final result in this study.

To address the specific questions, regarding BRCA1 and BRCA2, as mentioned in the manuscript only two patients had mutations, both being CHEK2 mutations, so no patients had a known BRCA mutation in this cohort. While we are happy to restate this explicitly, we also have concern that listing these results multiple times in different sections will make the manuscript burdensome to read. Family history was included as mentioned in the manuscript. Regarding treatment consistency, the only treatment factor of significance was receipt of neoadjuvant therapy, because the pathology of the excisional biopsy of the high-risk lesion was the endpoint of the study. We accounted for receipt of neoadjuvant therapy in our model. To assess the impact of specific neoadjuvant regimen when only 18 patients received it and only 2 of those 18 patients had lesion upgrade would not allow for any meaningful interpretation, which is why we limited our analysis to receipt of neoadjuvant systemic therapy.

Finally, regarding interaction terms, as this is a multinomial logistic regression with only five events of interest in the upgrade category, the assessment of interaction terms would not provide any meaningful results. If concurrent high-risk lesions were documented in large national databases such as SEER or NCDB, then such interaction terms would be very interesting, but in the context of a 65 patient retrospective study with 5 events, any result from such an analysis would not hold any meaning.

  1. The study included genetic testing results but did not specify the types of mutations found. Were there specific genetic mutations associated with a higher risk of lesion upgrade? Specifying the types of genetic mutations found and their association with lesion upgrade can provide more insight into which patients are at higher risk.

We thank reviewer #2 for their question regarding genetics. As described in the manuscript, there were only two patients with mutations, both CHEK2 mutations, so analysis of genetic mutations and of CHEK2 mutations are the same. From the original manuscript results 3.1, lines 139-142:

“There were 22 patients who underwent genetic testing, of whom only two had mutations identified, both of which were CHEK2 mutations. One of these two patients had an upgrade of her high-risk lesion site to invasive disease, and the other had persistence of her high-risk lesion.”

  1. What kind of follow-up care was provided to patients with high-risk lesions that did not undergo excision? The study does not specify the follow-up duration for patients. Long-term follow-up is important to understand the true upgrade risk of high-risk lesions. Including information on patient follow-up care is important to understand how these patients were monitored over time and to ensure any progression was detected early.

We thank reviewer #2 for their question regarding follow-up of high-risk lesions that were not excised. As described in the methods section, such patients did not meet criteria for inclusion in our study (lines 106-113).

The reason we did not include such patients is that they would not contribute to the analysis of our stated aim to identify factors that may help predict whether a lesion will upgrade or downgrade at time of excisional biopsy. There is no way of knowing with certainty that patients diagnosed with carcinoma do not harbor DCIS or invasive disease in their incidentally detected high risk lesion at the time of surgery if they do not undergo excision of the lesion. Imaging or serial core needle biopsy is inadequate to definitively rule out the presence of DCIS or malignancy. We hope that our study aim may be further addressed with larger future studies that may identify patient or lesion factors that may lead to a standardized protocol for monitoring and biopsy of these high-risk lesions as an alternative to surgical excision. A retrospective analysis of available data would otherwise be heavily biased by the specific follow-up received by the patient, as the more stringent and invasive the surveillance protocol, the higher the pretest probability that the patient will be upgraded.

  1. The study did not consider the cost-effectiveness of excising high-risk lesions. Did you evaluate if routine excision of these lesions is cost-effective compared to surveillance? Evaluating the cost-effectiveness of excising high-risk lesions versus surveillance is important because it can impact healthcare

We thank reviewer #2 for their question regarding cost-effectiveness of excising high-risk lesions.

Similar to the response to #9, the aim of this study was not to assess cost-effectiveness. In order to appropriately address cost-effectiveness, the outcomes of different treatment algorithms must be adequately delineated. While our study represents an assessment of the risk of upgrade of a high-risk lesion at the time of surgery for a primary invasive lesion, the risk of upgrade *after* surgery for a primary invasive lesion is a completely different topic. This would require a protocol to be followed that allows for comparison across patients so that results are not biased towards patients who undergo more aggressive surveillance and testing. We have no data to use for this purpose and to base a cost-effectiveness study on the results of a single institution retrospective study would have an inherent bias.

  1. The study did not perform a power analysis. Did the study have sufficient power to detect differences in upgrade rates between different high-risk lesions? Performing a power analysis is important to ensure that the study had enough participants to detect meaningful differences in upgrade rates, providing confidence in the results.

We thank reviewer #2 for their question regarding power analysis.

We do not conduct power analyses for retrospective studies as power analysis is valid for prospective studies, and the dangers and limitations of retrospective, post-hoc power analysis have been detailed by many different statisticians and practitioners across many different areas of research. Below is a short list of examples but we are happy to provide more if requested.

Nakagawa, S., & Foster, T. M. (2004). The case against retrospective statistical power analyses with an introduction to power analysis. Acta Ethologica, 7(2), 103–108. https://doi.org/10.1007/s10211-004-0095-z

Heckman MG, Davis JM 3rd, Crowson CS. Post Hoc Power Calculations: An Inappropriate Method for Interpreting the Findings of a Research Study. J Rheumatol. 2022 Aug;49(8):867-870. doi: 10.3899/jrheum.211115. Epub 2022 Feb 1. PMID: 35105710.

Lihshing Leigh Wang, Retrospective Statistical Power: Fallacies and Recommendations, Newborn and Infant Nursing Reviews, Volume 10, Issue 1, 2010, Pages 55-59, ISSN 1527-3369, https://doi.org/10.1053/j.nainr.2009.12.012.

Gilbert, Gregory E. and Susan K. Prion. “Making Sense of Methods and Measurement: The Danger of the Retrospective Power Analysis.” Clinical Simulation in Nursing 12 (2016): 303-304.

Gerard, Patrick D., David R. Smith, and Govinda Weerakkody. “Limits of Retrospective Power Analysis.” The Journal of Wildlife Management 62, no. 2 (1998): 801–7. https://doi.org/10.2307/3802357.

  1. The study did not discuss the impact of the interval between the diagnosis of high-risk lesions and invasive cancer. How did the time interval affect the upgrade rates? Discussing the impact of the time interval between diagnoses is important because it can influence the progression of high-risk lesions to cancer. Longer intervals might show different upgrade rates compared to shorter ones.

We thank reviewer #2 for their question regarding the duration between the diagnosis of invasive cancer and the high-risk lesion. To address this, we calculated the times between the diagnosis of the high-risk lesions and the invasive cancer and tested it in a univariate model, as well as the time between the diagnosis of the high-risk lesion and the concurrent surgery for the two lesions. Neither of these variables was significant in the model and so they were not passed onto the multivariate model (Results 3.3, lines 209-210)

  1. These questions and critiques should be added to the discussion section of the article to address potential limitations and provide a more comprehensive understanding of the findings. If there are specific sections where the data is missing, they should be added to ensure the study is thorough and well-rounded.

Here also are the main limitations of this retrospective cohort study that must be cited in the discussion. These limitations mean that the study's findings should be interpreted carefully and may not be broadly applicable. Please review this in the text:

  1. a) The study included only 65 patients. This is a small number, which makes it hard to apply the findings to a larger population.

We thank reviewer #2 for their question regarding the small sample size. We had already listed this as the primary limitation of the study, and aside from stating the small sample size there is little we are able to do about this. From the manuscript:

“The primary limitation of this study is the small sample size, both with respect to the overall cohort and when considering each specific type of high-risk lesion. This limits the generalizability of our findings for each type of lesion since several of them were grouped together and the lesion type was not predictive of lesion upgrade.” (Discussion, lines 295-298)

  1. b) The ethnic and racial diversity. The study did not thoroughly analyze the impact of ethnic and racial diversity on the upgrade rates. Different populations may have different risks.

We thank reviewer #2 for their question regarding ethnic and racial diversity.

As mentioned in the list of variables, race and ethnicity were curated and assessed in a univariate model. Similar to the other variables addressed above, since it was not significant it did not pass to the multivariate model (Results 3.3, lines 192-194).

  1. c) The study depends on past records, which might have missing or incorrect information.

We thank reviewer #2 for their question regarding the retrospective nature of the study. We modified the statement of limitations in the discussion section (Discussion lines 298-300):

“Other limitations include the retrospective, non-randomized, single-institution nature of the study and the inconsistent availability of data on the sizes of the high-risk lesions on pathology and on MRI.”

  1. d) Researchers cannot control how data was originally collected. This can affect the consistency and quality of the information.

We thank reviewer #2 for their question regarding the retrospective nature of the study. We modified the statement of limitations in the discussion section (discussion 298-300):

“Other limitations include the retrospective, non-randomized, single-institution nature of the study and the inconsistent availability of data on the sizes of the high-risk lesions on pathology and on MRI.”

  1. e) Looking back at past data means researchers cannot influence the study design or data collection, which might limit the study's reliability.

We thank reviewer #2 for their question regarding this study having the limitations of a retrospective study, which it is. . We have modified the statement of limitations in the discussion section (Discussion lines 298-300):

“Other limitations include the retrospective, non-randomized, single-institution nature of the study and the inconsistent availability of data on the sizes of the high-risk lesions on pathology and on MRI.”

  1. f) There could be selection bias because the study only includes patients who were treated at one specific center and met certain criteria. The study was conducted in a single center. This means the sample may not represent the broader population, limiting the generalizability of the results.

We thank reviewer #2 for their question regarding this study having the limitations of a single-institution study, which it is. We modified the statement of limitations in the discussion section (Discussion lines 298-300):

“Other limitations include the retrospective, non-randomized, single-institution nature of the study and the inconsistent availability of data on the sizes of the high-risk lesions on pathology and on MRI.”

  1. g) Only patients who underwent surgery were included. This could introduce selection bias because patients who did not have surgery might have different outcomes.

We thank reviewer #2 for their question regarding the inclusion of only patients who underwent surgery.

As previously described in question #9, such patients were not included because they do not contribute to the aim of the study. There is no way of knowing whether such patients would have had upgrade to DCIS or invasive disease if they did not undergo excisional biopsy. Hence our reason for excluding them. This is an inherent limitation of the study design. We edited a sentence of the limitations to reflect this and later questions by reviewer #2 in the discussion section (Discussion lines 298-300):

“Other limitations include the retrospective, non-randomized, single-institution nature of the study and the inconsistent availability of data on the sizes of the high-risk lesions on pathology and on MRI.”

  1. h) Inconsistent Definitions. The study does not clearly define what constitutes an "upgrade" in all cases, which can lead to inconsistent interpretation of results.

We thank reviewer #2 for their question regarding definition of an upgrade.

We removed the word “invasive” from the simple summary (line 19).

We also edited the following sentence in an attempt to make it plenty clear how an upgrade is defined (Methods 2.3, lines 119-121):

“Regression models were developed to identify preoperative risk factors associated with upgrade to malignancy (invasive carcinoma or ductal carcinoma in situ [DCIS]) or downgrade to no evidence of malignancy or high-risk lesion.”

  1. i) The results might not apply to all patients with similar conditions because the study was done in one hospital with specific patient demographics.

We thank reviewer #2 for their question regarding this study having the limitations of a single-institution study, which it is.

We edited a sentence in the limitations portion of the discussion to explicitly clarify this in case any readers are not familiar with the many limitations of retrospective analysis of prospectively maintained databases (Discussion lines 298-300):

“Other limitations include the retrospective, non-randomized, single-institution nature of the study and the inconsistent availability of data on the sizes of the high-risk lesions on pathology and on MRI.”

  1. j) Lack of randomization. As a retrospective study, there is no randomization. This can introduce various biases and affect the study’s validity.

We thank reviewer #2 for their question regarding lack of randomization.

We edited a sentence in the limitations portion of the discussion to explicitly clarify this in case any readers are not familiar with the many limitations of retrospective analysis of prospectively maintained databases (Discussion lines 298-300):

“Other limitations include the retrospective, non-randomized, single-institution nature of the study and the inconsistent availability of data on the sizes of the high-risk lesions on pathology and on MRI.”

  1. k) Patient Preferences: The study does not consider patient preferences or the psychological impact of knowing they have a high-risk lesion. This can influence decisions on treatment and follow-up.

We thank reviewer #2 for their question regarding psychological impact of high-risk lesions.

As previously stated, the aim of this study was to identify risk factors for upgrade at time of excision. The only treatment and follow up that psychological factors would affect is that of whether the patient chooses to forgo excisional biopsy. There is no way for us to assess this as explained in our response to question #9. Furthermore, as this study is retrospective in nature, we are unable to identify patient preferences or the psychological impact of them knowing they have a high-risk lesion. Contacting patients to acquire such information would be prone to significant recall bias. And would not contribute to our assessment of the aim of the study.

  1. l) Treatment Consistency: Variations in treatment protocols over the 15-year study period are not addressed. Consistent treatment approaches are important for reliable results.

We thank reviewer #2 for their question regarding treatment consistency.

As described in the response to question #7:

Regarding treatment consistency, the only treatment factor of significance was receipt of neoadjuvant therapy, because the pathology of the excisional biopsy of the high-risk lesion was the endpoint of the study. We accounted for receipt of neoadjuvant therapy in our model. To assess the impact of specific neoadjuvant regimen when only 18 patients received it and only 2 of those 18 patients had lesion upgrade would not allow for any meaningful interpretation, which is why we limited our analysis to receipt of neoadjuvant systemic therapy.

Round 2

Reviewer 2 Report

Comments and Suggestions for Authors

I'm pleased to let you know that after a careful review, your manuscript is now in excellent condition for publication. All of the earlier concerns have been successfully addressed by the revisions, greatly improving the paper's overall quality and clarity. Your research's breadth and depth are noteworthy for their value and depth, contributing notably to the field.

I appreciate your commitment to polishing the manuscript. I do not doubt that the academic community will value your work and that it will significantly advance current discussions and research in your area.